# Biological Contribution of Ornamental Plants for Improving Slope Stability along Urban and Suburban Areas

Alessandra Francini [1], Stefania Toscano [2], Daniela Romano [2,*], Francesco Ferrini [3] and Antonio Ferrante [4]

1 Institute of Life Sciences, Scuola Superiore Sant'Anna Pisa, Piazza Martiri della Libertà 33, 56127 Pisa, Italy; francini@santannapisa.it
2 Department of Agriculture, Food and Environment (Di3A), Università degli Studi di Catania, Via Valdisavoia 5, 95123 Catania, Italy; stefania.toscano@unict.it
3 Department of Agriculture, Food, Environment and Forestry (DAGRI), Università degli Studi di Firenze, Viale delle Idee, 50019 Sesto Fiorentino, Italy; francesco.ferrini@unifi.it
4 Department of Agricultural and Environmental Sciences, Università degli Studi di Milano, Via Celoria 2, 20133 Milan, Italy; antonio.ferrante@unimi.it
* Correspondence: dromano@unict.it

**Abstract:** Plants can reduce erosion during heavy raining periods and improve slope stability through their root morphology, development, biomass, and architecture. Heavy rains can increase erosion, becoming a danger for traffic and people who live around slopes. The control of slope stability is often required in urban and peri-urban environments, and for this reason ornamental species can be appropriately selected for a dual use, namely improving the aesthetical value of green areas along the urban and suburban roads and mitigating the erosion effects. The species used must have good tolerance to abiotic stresses, such as high and low temperature, drought, pollution and nutrient deficiency. Otherwise, their limited growth can reduce their beneficial effects. Ornamental plants that can be used for reducing the erosion of slopes must be in full growth during periods with a higher incidence of rains and must also be compatible with the temperature ranges in different seasons. These species can be also selected for their ability to avoid erosion and enhance the stability of slopes. In this review, the biological contribution of plants for improving slope stability has been reported and discussed with a special focus attention on the Mediterranean environment. Particular emphasis has been placed on root biomass changes and root growth parameters, considering their role as potential markers for selecting suitable plants to be used for enhancing slope stability. A brief description of planting on slopes and root growth has been also considered and discussed.

**Keywords:** root development; root morphology; abiotic stress; growth regulators; biostimulants; plant choice

## 1. Introduction

Urban and peri-urban green areas provide important ecosystem services for the quality of the urban environment such as air pollution mitigation, direct effects on local climate, noise abatement, stormwater management during rainy periods, carbon dioxide assimilation, oxygen supply, and recreational and social benefits [1]. Turfgrass, ornamental shrubs, and trees, can deliver different ecosystem services beyond their aesthetical contribution depending on the composition and biodiversity [2]. They are also used for improving the stability of slopes along roads and in urban areas [3]. The stability of slopes is mainly due to plants' root biomass, distribution, and architecture [4]. The current review has the objective of highlighting the biological contribution of ornamental plants to increasing slope stability. Through a review of the literature, the current work will explain how ornamental plants (with their habitus, growth, and roots systems) can prevent erosion and improve slope stability in urban and suburban areas.

Roots grow in soil and create an underground net able to reduce or avoid erosion and landslides that can be dangerous and induce severe damages in urban environments.

Plant density and biodiversity affect both the number and architecture of roots and their contribution to the slope stability through hydrological and mechanical processes [5]. Root morphology and growth are influenced by genetic background, soil characteristics, and climate conditions (i.e., prevailing winds) [5]. The root morphology, in terms of diameter or ramification, can enhance the soil held back and the shear-strength of the rooted soil; a higher root diameter has beneficial effects on soil stability, acting as strong underground grid. Moreover, roots can improve water retention during the raining period and create a drainage network that allows for soil water absorption, avoiding runoff and erosion [6]. The soil around the roots is hydrologically and mechanically more stable and the consequences of this are easier infiltration, better physical and chemical properties of the soil, and higher shear strength [7]. All of these factors can positively or negatively affect soil erosion [8]. Factors reducing the soil erosion are mainly represented by vegetation or physical barriers [9]. The effect of vegetation on erosion control can be dramatically observed in areas subjected to fire events, where the massive destruction of plants and their roots system cannot hold the soil in slope conditions with high incidence of landslides. Roots are living organs and are subjected to turnover. Therefore, dead roots generate empty channels useful for drainage [6,10].

Slope stability also depends on the roots' depth, uniformity, and distribution. Different plant species should be closely planted, and the selection of species should be conducted with regard to the root distribution in soil and their interactions, avoiding those that can have antagonist responses. Plants with deep root systems should be associated with species having superficial roots, providing a good root network at different depths [10]. The stability of soil in planted slopes depends on the interaction of roots of different species that can synergistically work or have antagonist effects. Some allelopathic compounds could reduce the efficacy of the plants to prevent erosion or landslides. Therefore, plant selection for slope greening must be accurately carried out. A wrong plant species combination can limit plant growth and benefits, and this can be also a disadvantage for slope stability [11].

However, plant leaf area, branch density and ramification can reduce the energy of precipitation (soil impact) of rain and runoffs. The combination of turfgrass, shrubs, and ornamental trees can simultaneously reduce the kinetic force of rain with multiple canopies at different heights and water run-off is slowed down by grass [12]. The reduced superficial run-off velocity increases soil water infiltration exploiting, the channels created by the roots of different species. In evergreens, the canopy can also reduce the snow accumulation on the slopes avoiding possible landslide events. Evergreen shrubs or trees can also remove the water from the soil through transpiration (even in winter) and by reducing landslides (even if low temperatures slow down plants' metabolism). Slopes subjected to landslides should be preferentially covered by evergreens, these being deciduous plants with inactive roots during winter, thereby possessing a lower stability efficacy in the rainiest season [13].

Biophysical properties can modify the contribution of roots in stabilizing slopes. Among Mediterranean trees and shrubs, for instance, Moresi et al. [14] found that the most resistant roots to breaking under tension were those of *Quercus cerris* L., while roots of *Ilex aquifolium* L. had the highest tensile strength among all shrub species. In cold or freezing environments, on the other hand, ornamental plants must be selected among those that are tolerant to low temperatures and evergreens. Analysing the plant-root-reinforced shallow slopes, Tsige et al. [15] observed that the effect of vegetation on slopes increased when the spacing between plants decreased, and that the slope angle modification in combination of plant roots had a relevant influence. Among the analysed species, *Salix subserrata* Willd. was the most promising plant species for slope stabilization, due to the effect of its better root mechanical properties.

However, it must be highlighted that the positive effects in the urban environment can be obtained if green space management is regularly carried out. Therefore, species must be selected considering technical parameters but also the municipal budget for urban green areas management [16].

Information related to the urban environment and to the ornamental plants used in the green areas is limited if compared to that of the natural environment, while it could be a support for city managers in taking decision about slope management. In the review, attention was therefore given to providing useful information for this purpose, with particular emphasis on the Mediterranean environment where extensive periods of water stress in summer and rainfall occurrences through intense precipitations that can turn into severe flash floods and floods can accentuate the problems of stability of the slopes.

### 1.1. Root Growth under Slopes Increase

Roots and canopies are strictly connected, and their relations dynamically change in terms of biomass, architecture, and organization during growth and according to the season conditions. Roots' distribution in soil follows the nutrients and water availability. Roots represent the transport system for nutrients and water and connect soil with leaves where the most part of metabolic processes take place. Each species has a specific roots architecture [17] and distribution in the soil (Figure 1). The genetic background of the species defines the potential root growth and distribution in the soil (i.e., depth, spreading, density) (Figure 1A–E). Roots mainly grow close and parallel to the soil surface and can strengthen the soil by intensifying the in-plane tensile strength of the rooted soil [5], while roots growing perpendicular to the soil surface strengthen the soil by improving shear-strength of the rooted soil mass on the sheared surface.

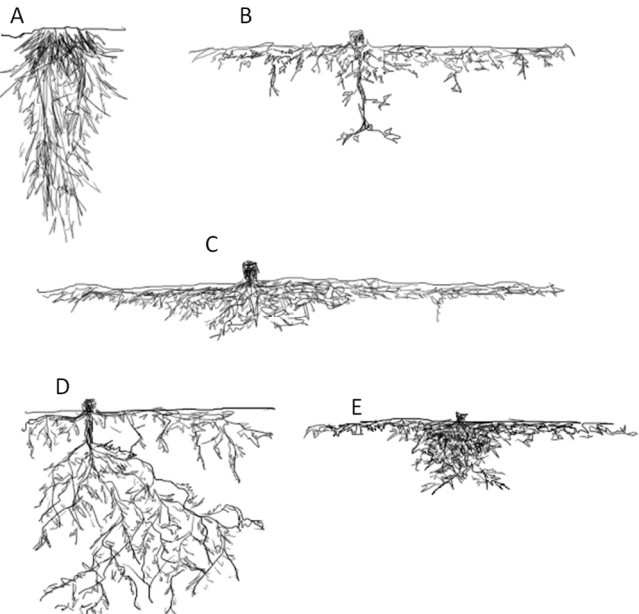

**Figure 1.** Different root systems development. (**A**) Tuft and deep root systems, (**B**) taproot root system, (**C**) superficial root system, (**D**) taproot and horizontal lateral roots, (**E**) heart-shaped root architecture. Redrawn by the authors from Ghestem et al. [5].

Roots of ornamental plants growing in slope conditions, in addition to nutrient and water transport, must guarantee plants' biomechanical stability and indirectly contribute to the slope stability. The increase of the slope induces different root responses and growth modifications. Mechanical stresses such as wind, rain, and gravitational force in sloping conditions influence root growth and their distribution, as mentioned above. As a consequence, the same species grown in different slope conditions can have different roots system that function to increase plant stability. Root growth under different slope degree has not been sufficiently detailed and this knowledge gap can represent limitations for plant species selection for practical applications. Further studies should be carried

out for elucidating how roots can balance the mechanical stability and their physiological functions.

Plant growth in slope conditions must follow the positive and negative gravitropic responses [18]. In a sloped soil, the aerial part and roots tend to align vertically during growth (Figure 2A). It means that plants in slope conditions have canopy and roots distribution that are not symmetric but this growth conditions could reduce the beneficial effects of plants for preventing landslides (Figure 2B).

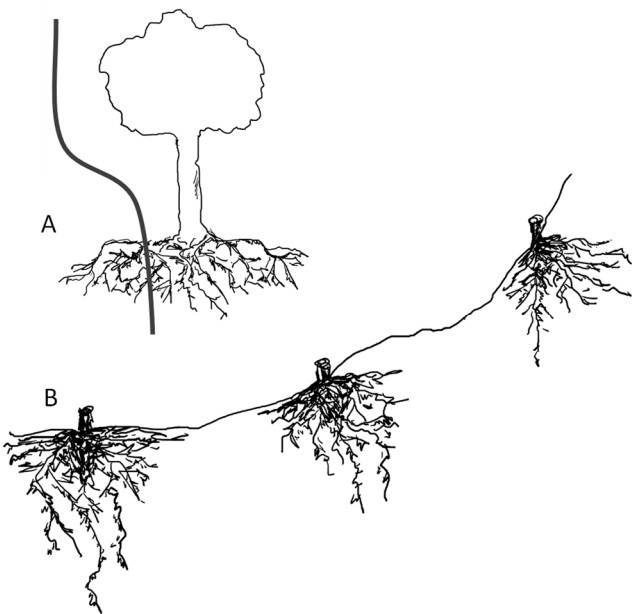

**Figure 2.** Plants grow in response to gravitropic stimuli: (**A**) the line represents the plant growth response to gravity. In slope conditions, the plants try to grow in a vertical position, reducing the angle between the ax of the trunk and the soil in slope, similarly, roots respond to gravity underground. Roots growth follows negative gravitropic response and aerial part positive response; (**B**) the increase of the slope influences in roots growth, development, and distribution which are a resultant of the gravitropic response and environmental conditions (original drawing).

As mentioned, root stability function is primarily determined by the biomass. However, root distribution in soil also plays an important role [19]. Unfortunately, the study of root distribution in soil is very complicated and in vivo monitoring of root growth and architecture distribution cannot be easily performed without having a perturbation effect on the roots system. On the contrary, invasive root analysis can lead to unreal results, especially regarding thinner apical roots. Plants located in slope conditions are more vulnerable to climatic events with potential danger for the closer buildings or roads [20]. Herbaceous plants combined with shrubs and trees can increase the roughness of the surface and enhance water infiltration, but higher water content in soil can increase the pressure and the weight and the susceptibility to landslides. The role of roots of ornamental plants in slope stability needs further investigations and multidisciplinary approaches are required for understanding the biological, hydraulic, and mechanical related aspects. Using biological solutions can have several applications for slope stabilization, but the lack of information regarding species behavior in different environments, especially in urban environments, represents one of the most significant limitations.

### 1.2. Root Morphology and Slope Stabilization

Ornamental plants should be selected considering the root morphology and development in relation to physical (sandy, clay, etc.) and chemical (mainly pH and salinity) soil proprieties and depth. For rocky slopes along urban areas with limited soil availability

ornamental plants with superficial (Figure 1C) or heart-shaped (Figure 1E) root systems should be used. On the contrary, for slopes with adequate soil depth plants with deeper root systems that can ensure higher slope stability can be selected [21]. The response of different root systems under different slope conditions should be further investigated for improving slopes stability through greening with ornamental plants.

## 2. Root Physiology and Development under Abiotic Stress

The root system is an integral plant organ involved in the acquisition of nutrients and water, the synthesis of plant hormones, organic acids, and amino acids, and it is necessary to ensure the anchorage of plants [22,23]. It also plays a fundamental role in maintaining cellular homeostasis under normal growth conditions and in plant-to-plant communication. Through the roots, plants absorb water and nutrients from the soil and transfer them to the aerial part. During stressful conditions, this equilibrium is modified, and the roots must implement structural and functional changes [24]. Root morphology and physiology are closely associated with the growth and development of aboveground plant material. However, it is known that the degree of the responses of roots to abiotic stresses may vary considerably within a family, a genus, and even a species [25].

In the presence of different abiotic stresses, the root system is modified due to presence of phytohormones that regulate this process [26]. The root is the initial part affected by abiotic stresses, and its morphological and physiological characteristics are closely correlated with plant resistance [27] or tolerance.

Different abiotic stresses affect the plants, and the drought stress is among the most important. In order to overcome drought stress conditions, plants modify the root system morphology and activate different physiological and biochemical processes [28,29].

A first response of the plant to an abiotic stress is the modification of the root biomass. In drought conditions, the roots, as the principal organ for water and nutrient uptake, play an important role in the plant drought tolerance [30]. *Penstemon barbatus* (Cav.) Roth, for instance, was able to tolerate drought by increasing root biomass and reducing stomatal conductance [31]. Similar results were observed in sunflowers (*Helianthus annuus* L.) [32] and *Catharanthus roseus* (L.) G.Don [33–35].

Even in presence of salt stress, the root biomass is modified. The osmotic stress caused by the salts present in the root environment induces a decrease in soil water potential at the root surface and, as a consequence, a difficulty for water uptake by the plants (decreases in leaf osmotic potential and leaf water potential) [36–38]. Fornes et al. [39] observed in three ornamental species (*Calceolaria* × *herbeohybrida* Voss, *Calendula officinalis* L., and *Petunia* hybr.) that root growth was reduced by salinity. During salt stress conditions, root dry biomass is an important parameter, because the higher root growth allows higher water and nutrient uptake to take place, favoring the accumulation of toxic ions in roots, in particular $Na^+$, thus diminishing its negative effects on shoot growth [40].

Drought and salt stresses can indirectly modulate root system architecture since they can produce unfavourable changes in the soil nutrient composition and distribution, soil density and compaction, and the type of soil particles [41]. Root growth is then deeply influenced by the availability and by the quality of water in the soil; the root system is the first to perceive the stress signs due to drought and salinity. Water scarcity inhibits the growth and development of the whole plant in numerous important species, while the root system, which is more tolerant than the aerial part, continues to grow even in the presence of low water potentials [42].

By increasing their root system, plants are able to explore the deeper layers to obtain water. An increase in the biomass ratio between roots and shoots (R/S ratio) under drought stress confirms this statement [43]. When the plant is affected by extreme soil drought, the regulation capacity, through asymmetric growth approach, may be also lost abruptly [43].

Due to drought stress, the plants modify the R/S ratio, for reducing water consumption and increase water absorption [44,45]. The increase in R/S ratio was observed in different species: *Lonicera implexa* Aiton [46–48], *Lupinus havardii* S.Watson [49], *Myrtus*

*communis* L. [50], *Nerium oleander* L. [51,52], *Opuntia ficus-indica* (L.) Mill. and *O. robusta* J.C. Wendl. [53], *Rhamnus alaternus* L. [54], *Rosmarinus officinalis* L. [55], two rose rootstocks (*Rosa multiflora* Thunb. and *R.* × *odorata* (hort. ex Andrews) Sweet) [56], *Sambucus mexicana* C.Presl ex DC. [57], *Silene vulgaris* (Moench) Garcke [58,59], in different shrubs of Mediterranean basin [60], and in *Viburnum tinus* L. [61]. Tribulato et al. [61] found an increase in root dry biomass (a higher R/S ratio) in *V. tinus* plants subjected to a severe water stress condition. This adaptation would allow to overcome the transplant condition and water shortage [62,63]. It is proved that the increasing R/S ratio is considered one of the avoidance mechanisms enabling plants to maximize the water uptake under drought stress condition [64].

In presence of drought stress, the plants need to maintain a greater root surface, while in salt stress conditions in some cases they can even reduce the root surface to limit the accumulation of toxic ions in the shoot, inducing in both situations a different distribution of the roots [65–67]. This can reduce water depletion around roots for minimising resistance to water transport to the root system [68], and modifies the water use efficiency that is improved, as demonstrated by Fernández et al. [69] on *Phillyrea angustifolia* L. Gomez-Bellot et al. [70] and by Álvarez and Sánchez-Blanco [71] who reported an increase in the R/S ratio in *Euonymus japonicus* Thunb. and *Callistemon citrinus* (Curtis) Skeels plants under moderate salt stress (EC 4 dS m$^{-1}$). Under salt stress condition this modification is frequently observed in plants [72]. Cirillo et al. [73] found unchanged R/S ratio in *Callistemon citrinus* and *Viburnum lucidum* L. plants subjected to salt stress; one interpretation of the unchanged R/S ratio may be a greater severity of NaCl stress (200 mmol NaCl).

During drought conditions not only the R/S ratio is modified but also other root characteristics such as root length, fresh weight (FW), dry weight (DW), diameter and surface area, deep rooting and cortex thickness and behaviours (i.e., root turnover, metacutisation, hardening, and hydraulic conductivity) can be influenced [68,74].

In *Callistemon* plants, Álvarez et al. [75] observed that water deficit increased the percentage of fine roots and decreased those with a diameter higher than 0.5 mm. In general, stressed plants showed a reduced root volume, although root dry weight was not modified, with the result that root density increased [34,75]. The root diameter increases with depth, and it is greatly linked to the uptake of water in deep soil layers [76].

Another factor that plays an important role in the tolerance of the drought stress is the hydrotropism [77,78]. Takahashi et al., [79] in a study about *Arabidopsis* and radish demonstrated that a gradient of moisture determined by water stress induces an immediate degradation of amyloplasts in the columella cells of plant roots, producing less response to gravity and increasing the hydrotropism.

A higher percentage of fine roots, able to penetrate the smallest soil pores, presumably optimises the exploratory abilities of the root system, and may play an important role for the survival of plants during drought stress [80]. Instead, in *Myrtus communis* L. and *Nerium oleander* L. plants, the percentage of thick roots increased, and the percentage of medium and fine roots was reduced following drought stress [50,51]. Different authors reported an increase in root diameter in different species (*Picea* sp., *Pinus banksiana* Lamb., *Portulaca oleracea* L.) in response to salt stress [74,81]. The higher robustness and accumulation of reserves, observed in these plants, could be linked to a higher root density [74,75,82].

Several studies have reported the association of the length, volume, and density of roots in crop species with drought tolerance [83–86]. Drought stress decreased the root length in *Abelmoschus esculentus* (L.) Moench [87], *Albizzia* seedlings [88], *Eucalyptus microtheca* F. Muell. seedlings [89], *Nerium oleander* L. [51], *Rhamnus alaternus* L. [67]. The opposite effects of drought stress on root length in other species agreed with the results of Chyliński et al. [90] on geranium (*Pelargonium hortorum* L. H. Bailey) and impatiens (*Impatiens walleriana* Hook) also reported by Shober et al. [91] on *Viburnum odoratissimum* Ker Gawl. and by Franco et al., [59] on *Silene vulgaris* (Moench) Garcke.

The development of lateral roots is inhibited in conditions of water stress, while the induction of new roots does not change and the development of primary roots increases

due to the hydrotropism that are under the control of ABA [92]. However, this is not always confirmed. Indeed, in a study conducted by He et al. [93] on *Camellia oleifera* Abel, subject to drought stress, plants at the end of the experiment did not show a significantly decrease in the number of lateral roots. The root-crown and root-plant ratios indicated that in drought conditions the plant gives priority to the normal development of the root system, by maintaining the contact area with the soil, thus obtaining the necessary water [94]. This has also been observed in other studies [95].

Hardening of roots, measured by increasing of brown root percentage, is frequent in drought-stressed plants [68]. Plants of *Limonium cossonianum* Kuntze [96], *Lotus creticus* L. [47] and *Silene vulgaris* (Moench) Garcke plants [59], with limited water availability, showed variation of root colour, from white to brown, that is linked to the suberisation of the exodermins and it is an index of metacutisation process. This was also found in *Rosmarinus officinalis* L. by Sánchez-Blanco et al. [55] and in *Nerium oleander* L. by Bañón et al. [51]. In two trials conducted by Franco et al. [59] about *Silene vulgaris* subjected to drought stress an increase in cortex thickness:root radius (C:R) ratio was observed improving the resistance to dehydration. These changes could increase the resistance of *Silene vulgaris* seedlings at the drought conditions.

On the contrary, flooding or waterlogging represent extreme conditions for plants and roots as the first target of waterlogging stress in plants. In urban environments, severe soil compactions and limited drainage can induce waterlogging, thus hypoxia and anoxia effects on plants. Soil waterlogging has in fact been identified as one of the main abiotic stresses in urban areas; the constraints imposed on the root have marked effects on the growth and development of plants [97,98]. Waterlogging inhibits respiration in the root, due to an insufficient supply of oxygen [99]. Hypoxia is the main stress factor in waterlogging conditions [100] and the primary effect of soil flooding is to slow down the oxygen transfer to the roots. This limits their aerobic respiration and dramatically depresses their metabolism. In addition, if tissues are hypoxic, the aerobic energy is reduced and the functional relationships between roots and shoots are compromised [101,102].

The development of adventitious roots is stimulated by the increase of ethylene production in the shoots of the trees or for effect of external increasing of the compound in the soil solution [103]. Formation of adventitious roots in response to ethylene has been considered a major adaptive mechanism of wetland plants to root damage caused by waterlogging stress [104]. Adventitious roots emerge and grow horizontally close to the water surface, and they are connected to the stem close to the site of aerenchyma formation [105]. Hence, adventitious roots can facilitate oxygen capture of submerged tissues alleviating the hypoxic conditions and contributing to the recovery and maintenance of aerobic respiration in waterlogged seedlings [106,107].

Under salt stress conditions, the root mitochondrial electron transport might be disrupted, promoting $O_2$ accumulation in a manner similar to that from hypoxia conditions [108]. Tolerant species exhibit several physiological and biochemical modifications including quasi-dormancy of shoot tissues, stomatal closure, elongation of submerged stems, and the formation of aerenchyma in existing root tissue or development of new nodal roots at the stem base [101].

The density of the xylematic vessels is one indicator of plant capacity to absorb and transport water in the roots. In drought conditions, a higher vessel density could increase the tolerance. This was observed in a study on *Lotus creticus* L. subjected to drought stress [48].

In northern countries, the abiotic stress predominant during winter is due to cold or freezing. The cold can damage the cell membrane and the severity of damage can reach the leaves or branches of evergreens. In slope conditions, the plants used for enhancing the stability must have a high tolerance to freezing temperatures, especially at the root level. The ice formation in the soil can induce roots damage or death. In these conditions, landslides can occur with the de-icing when the connection between roots and soil is

lessened. In cold environments, it is very important to select winter herbaceous species and evergreen shrubs or trees.

As above described, certain abiotic stresses could stimulate root development and indirectly improve slope stability. The green areas of slopes are not subjected to constant management, and therefore the abiotic stresses such as drought represent common and frequent stress in peri-urban greens. Based on the predominant abiotic stress, suitable ornamental plants should be selected so that the response of the plants will probably be an increase of the root systems that improves the soil stability.

### 3. Use of Plant Growth Regulators and Biostimulants for Increasing Root Biomass

In slope conditions, transplanted plants rapidly need to develop their roots to ensure a good anchorage with soil so that they can soon reach stability. The root development can be enhanced with plant growth regulators such as cytokinins, auxins, or using plant biostimulants. The increase of roots length and biomass can be a response of a plant hormones equilibrium that is influenced by the external stimuli. Beside plant growth regulators, biostimulants can be also used as agronomic tools for stimulating root development and biomass accumulation. Research on biostimulants and roots demonstrated that several of these products can be effective for root formation and growth. Biostimulants can be derived from different raw materials, seaweeds, plant extracts (botanicals), inorganic compounds, beneficial fungi, and bacteria and are commonly used for increasing plant growth and abiotic stress tolerance [109,110]. Biostimulant applications in nursery or after transplant can enhance root formation.

The main application of these products is for production purpose in horticultural crops, but also in urban environments [111]. However, their environmental impact is low since most of them have organic nature. Among the plant extract, there are only few published works. Willow bark extract formation was effective in the development of adventitious roots and root branching in lavender and chrysanthemum [112]. Plant growth-promoting rhizobacteria (PGPR) applications can also improve root biomass and functionality and several positive results have been reported for flower and ornamental plants [113]. In *Eucalyptus* clones (hybrid *Eucalyptus grandis* W. Hill × *E. urophylla* S.T. Blake) the application of *Aspergillus flavipes* (ATCC®16814™) has been used as a novel biostimulant for rooting-enhancement, in terms of biomass and root length [114]. The positive action of PGPR is also due to the induction of auxins or cytokinins by the roots. The fast root development after transplant can rapidly cover the soil and in slope conditions is very important for reduce erosion.

The nursery phase influences plant development after transplant and also the root systems. In *Lotus creticus* L. subsp. *cytisoides* (L.) Arcang., the irrigation two days/week instead of six days/week determined a greater root length: shoot length ratio and a higher percentage of brown roots more favourable to tolerate transplant stress [115]. Also, treatments with arbuscular mycorrhizal (AM) in the nursery phase can improve the root system architecture and resistance to drought in *Pistacia lentiscus* L. [116].

Biostimulant fungi based such as *Trichoderma* increase root growth and nutrient uptake by the induction of auxin biosynthesis [117]. In *Impatiens walleriana* Hook. f. plants treated with *Trichoderma* isolates showed longer roots and higher roots dry weight than control and comparable with commercial indole-3 butyric acid (IBA). These studies demonstrated that different isolates have different efficacy and appropriate species should be used for roots induction and development [118]. Mexican petunia (*Ruellia brittoniana* Leonard) treated with humic acids, amino acids, and active dry yeast showed an increase in root length and weight, indicating the potential role of these compounds in roots development [119].

Seaweed extracts obtained from *Ascophyllum nodosum* (L.) Le Jolis applied to *Passiflora actinia* Hook. increased 10% of rooting with 40% seaweed extract [120]. The rooting stimulation of a commercial biostimulant-based seaweed extracts has been also demonstrated in woody cuttings of *Camellia japonica* L. [121]. Analogous results were also observed cutting of old rose cultivars [122,123]. These results were observed at nursery using cuttings; pre-

or post-transplant application should be also studied to understand if biostimulants could be applied as application for inducing rapid root development and reaching a rapid vegetation covering of slopes. Moreover, biostimulants could be also applied during the green management on slopes. These products stimulating the root development can rapidly enforce the integration of vegetation with soil and increase the slope stability.

## 4. Use of Ornamental Plants for Slope Greening

Ornamental plants are often valued only for their visual aspect. For this reason, the concept of ornamental plants is frequently used in its broadest sense to include plants that are grown for decorative purposes such as in the case of gardens, home gardens, landscape design projects, squares, parks, street trees, indoor plants, and cut flowers [124]. Recently, very widespread ecological requests have determined that ornamental plants are not only beautiful, but that any plant able to improve the environment and the quality of our lives [125] by providing ecosystem services is valuable. Ornamental plants can be adopted to restore degraded landscapes, and in particular to control erosion. In consideration of the numerous green area typologies and the breadth of the meaning of 'ornamental', the number of species that can be used is extensive [29]. The wide number of ornamental or potentially ornamental species enhances the possibility of identifying genotypes that are able to cope with the different conditions where these plants can be used.

However, it is also important to evaluate the geographical distribution of the different plant species during selection. It is advisable to avoid exotic species that can cause invasiveness problems. Soil erosion is a typical environmental problem that inflicts numerous and serious damages in agricultural cultivations as well as in natural ecosystems. In particular, erosion reduces the water-holding capacity of plants due to rapid water runoff and reduces soil organic matter [12], which can also affect green areas (especially those realized in slope surfaces). The key role of plant cover in controlling water erosion is widely accepted. According to Naylor et al. [126], the effects of vegetation on soil can be divided into two major categories: bioprotection, by reducing water runoff [12], and bioconstruction, by increasing water infiltration into the soil matrix [127].

Plants with their roots fix the soil [128] and with their canopy reduce the energy of raindrops [12]. The way the plants are arranged along the slopes can decrease the sediment runoff [129] (resulting from superficial down slope transport of soil particles) [7,130].

Gyssels et al. [130] stated that the aboveground vegetative cover was the most important factor to splash and interrill erosion processes; the roots were as important as aboveground vegetation cover for rill and gully erosion processes. The relative contribution of roots to runoff and soil loss reduction varied with vegetation types. Roots conserve soil or increasing infiltration, thus reducing runoff and soil loss [131], or improving soil properties by increasing soil organic matter levels, enhancing the quantity of soil stable aggregates and stabilizing soil layer structures [132,133].

Unfortunately, information on root characteristics of ornamental plants and their effects on the topsoil resistance to concentrated flow erosion is lacking. Roots influence the properties of the soil, such as infiltration rate, aggregate stability, moisture content, shear strength, and organic matter content, all of which control soil erosion rates to various degrees. The presence of roots also increases soil roughness, improving the capacity for water infiltration and reducing surface runoff velocity [134].

The impact of herbaceous and woody plants on soil erosion is crucial and different according to the species. Since ornamental plants can be both herbaceous and woody, it is possible to count on both effects against soil erosion. Perennial grasses provide year-round soil cover and reduce water runoff and sediment loss and promote soil-development processes by enhancing soil organic matter, soil structure and soil water and nutrient-holding capacity. Dense root architecture and vegetative cover on soil surface can reduce soil erosion. Woody plants reduce water erosion by improving water infiltration, reducing the negative effects of droplets, intercepting rain and snow and stabilizing the soil through roots and leaves.

In the semi-arid Mediterranean region, where water erosion is particularly severe, different experimental studies on the influence of the native vegetation on erosion have quantified soil loss and runoff under woodlands or shrublands comprising a mixture of plant species [135–137]. All of these studies have shown that typical Mediterranean shrubland vegetation is very efficient in reducing water erosion, also under extreme torrential simulated rainfalls [138].

In general, under similar climatic and topographic conditions, shrubs are most efficient in reducing runoff and sediment levels, followed by herbaceous plants and trees [133]. The coverage rate plays a similarly important role as vegetation type in affecting runoff and soil loss; the effects of vegetation types and coverage rates on runoff and soil loss are related.

Not all grass species, despite their many fine roots, appeared to have strong roots [139], and so species choice is very important in reducing erosion. The choice of suitable plant species depends on the context. De Baets et al. [140] used four criteria to evaluate the capacity of different species to control erosion, i.e., plants having: (i) a high potential to prevent incision by concentrated flow erosion, (ii) the potential to improve slope stability, (iii) the potential to resist bending by water flow, and (iv) the ability to trap sediments and organic debris. The scores for these indicators were represented on amoeba diagrams, indicating the strengths and the weaknesses of plant traits, in relation to erosion control. The scoring of plants on these criteria was based on a multi-criteria analysis. In the species choice, the plant tolerance to abiotic stress, and in particular to drought gains relevance, especially in the Mediterranean area.

In an experiment Bochet et al. [141] analysed the relative efficiency in reducing water erosion on slopes of three representative species of the Mediterranean vegetation, that showed different plant morphologies (*Rosmarinus officinalis* L., *Stipa tenacissima* L., and *Anthyllis cytisoides* L.). The results showed that the three species differently reduced runoff and soil loss. *Stipa* plants, characterized by dense canopy, counteracted rainfall erosivity, reducing splash erosion. *Rosmarinus*, in addition to the canopy effects, improves the topsoil structure by means of incorporation of organic matter. The litter cover seems to be very important in erosion control. *Anthyllis*, that is a deciduous shrub, give little protection against the impact of rain on soil surface as compared to a bare surface.

Burylo et al. [142] carried out an experiment to investigate the effect of the root systems of three species [*Robinia pseudoacacia* L. (tree), *Pinus nigra* var. *austriaca* (Höss) Badoux (tree) and *Achnatherum calamagrostis* (L.) P. Beauv. (grass)], on concentrated flow erosion rates. Ten functional traits, related to plant morphological and biomechanical features, were measured. Analyses were conducted to identify traits that cause plant root effects on erosion control. Erosion rates were lowest for samples of *R. pseudoacacia*, intermediate in *A. calamagrostis* and highest in *P. nigra* var. *austriaca*. The study also highlighted the role of fine roots in reducing erosion rates.

To compare the contribution in erosion control, five fern natives of southern China, namely, *Blechnum orientale* L., *Cyclosorus parasiticus* (L.) Farw., *Dicranopteris pedata* (Houtt.) Nakaike, *Nephrolepis auriculata* Trimen, and *Pteris vittata* L., were selected. The leaf area index, root area ratio and root density were significantly correlated with erosion-reducing potential. Among the species, *N. auriculata* performed better the other species by showing higher values of the determined plant traits [143].

Over the species choice (Table 1), the relationships between vegetation structural attributes (spatial pattern, functional diversity), soil surface properties (crust, stone, plant, and ground cover, and particle size distribution) and hillslope hydrologic functioning have been kept in consideration [144]. Since a typical landscape is a blend of species (herbs, grasses, shrubs, trees, etc.), it is possible to organize the plant arrangement to obtain the best result in erosion control to take advantage of plant and root characteristics.

**Table 1.** Species with ornamental value [1] suitable for erosion control.

| Species | Family | Plant Habitus | References |
|---|---|---|---|
| *Amorpha fruticosa* L. | Leguminosae | Shrub | [145] |
| *Anthyllis cytisoides* L. | Leguminosae | Shrub | [128,139,140,146] |
| *Artemisia vulgaris* L. | Compositae | Herb | [146] |
| *Atriplex halimus* L. | Amaranthaceae | Shrub | [128,139,140,147] |
| *Carpobrotus edulis* (L.) N.E.Br. | Aizoaceae | Succulent | [148] |
| *Comptonia peregrina* (L.) J. M. Coult. | Myricaceae | Shrub | [149] |
| *Dorycnium pentaphyllum* Scop. | Leguminosae | Shrub | [128,146] |
| *Hedera helix* L. | Araliaceae | Climber | [148] |
| *Hippophae rhamnoides* L. | Rhamnaceae | Shrub | [150] |
| *Lantana montevidensis* (Spreng.) Briq. | Verbenaceae | Shrub | [148] |
| *Lavandula lanata* L. | Lamiaceae | Shrub | [7] |
| *Limonium supinum* (Girard) Pignatti | Plumbaginaceae | Herb | [139,140] |
| *Lonicera japonica* Thunb. 'Repens' | Caprifoliaceae | Climber | [148] |
| *Medicago arborea* L. | Leguminosae | Shrub | [151] |
| *Myoporum parvifolium* R. Br. 'Prostratus' | Scrophulariaceae | Shrub | [148] |
| *Nephrolepis auriculata* (L.) Trimen | Nephrolepidaceae | Fern | [143] |
| *Nerium oleander* L. | Apocynaceae | Shrub | [128,140,146] |
| *Opuntia ficus-indica* (L.) Miller f. *amyclaea* and f. *elongata* | Cactaceae | Succulent | [152] |
| *Origanum bastetanum* L. | Lamiaceae | Herb | [7] |
| *Origanum vulgare* L. | Lamiaceae | Herb | [147] |
| *Psolarea bituminosa* L. | Leguminosae | Herb | [151] |
| *Putoria calabrica* (L.) DC. | Rubiaceae | Shrublet | [153] |
| *Retama shaerocarpa* (L.) Boiss | Leguminosae | Shrub | [128,139,140,146] |
| *Robinia pseudoacacia* L. | Leguminosae | Tree | [142] |
| *Rosa abyssinica* R. Br. ex Lindl. | Rosaceae | Shrub | [153] |
| *Rosmarinus officinalis* L. | Lamiaceae | Shrub | [128,140,141,146,153,154] |
| *Rosmarinus officinalis* L. 'Prostratus' | Lamiaceae | Shrub | [148] |
| *Salsola genistoides* Juss. ex Poir. | Amaranthaceae | Shrub | [128,139,140,146] |
| *Salvia lavandulifolia* Vahl | Lamiaceae | Shrub | [7] |
| *Santolina rosmarinifolia* L. | Compositae | Shrub | [7] |
| *Senecio jacobaea* L. | Compositae | Herb | [147] |
| *Tamarix canariensis* Willd. | Tamaricaceae | Tree | [128,139,140,146] |
| *Tanacetum vulgare* L. | Compositae | Herb | [147] |
| *Tephrosia vogelii* Hook. f. | Leguminosae | Tree | [155] |
| *Vinca major* L. | Apocynaceae | Herb | [148] |

[1] Species of Poaceae family, often successfully used for erosion control, are not considered.

Berendse et al. [156], analysing the effect of the use of four diversity treatments (one, two, four, and eight species commonly found in grassland), found that plant species diversity has important effects on the erosion resistance of slopes. The loss of species diversity, in fact, reduces the erosion resistance. They concluded that the presence of diverse plant communities on slopes are essential to minimize soil erosion.

In the US, highway departments request wildflowers for erosion control. On roadsides, wildflowers ensured a source of colour as well as erosion control [157]. Effects of wildflowers are linked to their biodiversity that helps to reduce soil erosion [158]. Roads have a great impact on the environment, habitat fragmentation, soil erosion, edge effects, and pollution. In order to reduce such impacts, native plants, naturally occurring in roadside vegetation and well adapted to those conditions, provide highly effective mixes for revegetation [159–161].

García-Fayos and Bochet [161], analysing climate change consequences and the increase in soil erosion, found that high plant species biodiversity and plant cover are negatively influenced by climate change and soil erosion, which negatively influences soil resistance to erosion, nutrient content, and water holding capacity. They also reported that plant species diversity weakly correlates with plant cover, but strongly with soil characteristics related to fertility, water holding capacity, and resistance to erosion.

In the experiments, different species were analysed to identify important plant traits that influence the hydraulic roughness to contrast erosion. The results indicated the stronger effect of density-weighted traits, demonstrating that communities with the best trade-offs among stem density, diameter, and leaf area are the key to mitigate soil erosion. For these reason, herbaceous ecosystems could play an important role in soil erosion mitigation [147]. Herbaceous vegetation, in fact, was more efficient than trees in improving aggregate stability [162]. In crop trees of *Vernicia fordii* (Hemsl.) Airy Shaw, aggregate stability was improved in the presence of herbaceous *Artemisia codonocephala* Diels. Mixtures of different plant functional types, typical of landscape arrangement, would improve soil conservation on slopes, by reducing both surface water erosion and shallow substrate mass movement [162]. The combination with nitrogen-fixing species will also be useful for providing this element for improving roots growth and development.

## 5. Limitations and Research Gaps in the Use of Ornamental Plants for Improving Slope Stability

The use of plants represents a long-term solution for preventing landslides in slope soils. Nevertheless, there can be some limitations that can be summarized as follows:

- *Slow roots establishment and stability*: the contribution of roots to stability increases with the roots' development and establishment. The highest stability is achieved when the species used reach maturity and the roots network is well integrated with the soil. Therefore, the stability of the slope is not immediately obtained (a limitation to be considered);
- *Unpredictable environmental effects*: plants development depends on environmental parameters and unpredicted events can reduce the efficacy of plants used in slope stabilization. These can also include soil-borne diseases and the limitation of the use of specific agrochemicals can represent an important limit in the maintenance of plants health.
- *High costs and regularity of maintenance*: public urban areas are under municipality maintenance and the lack of funds or of regularity in the management can compromise the benefit of the plants on the stability of the slopes;
- *Minimum of soil for plants growth*: the use of plants as slope stabilizers can be a solution if there is a minimum of soil for roots development and for a better anchorage that can harness the soil itself avoiding landslides.

The research gaps in the use of ornamental species for slope stabilization are represented by the limited case studies. Research activities should be focused from plant material propagation to agronomic management in the early stage of plant growth after transplant.

At the nursery level, appropriate strategies should be adopted to increase the biomass and functionality of the roots to enhance their development after transplants. Agronomic strategies should be evaluated for increasing roots development using PGPR or biostimulants. A wide range of species combinations should be studied in different urban environments with different soil and different slopes. In particular, further studies are required for evaluating the effect of abiotic stresses on the ability of plants to prevent landslides. Limited information is available for the effect of cold or freezing temperature on roots metabolism and their potential roles as slope stabilizers.

Moreover, the stability of vegetated areas should be mechanically determined for providing objective results. Since plants are living organisms, the stability of slopes should be studied over a long period and the weakness associated with the age of the vegetated area should be identified.

## 6. Conclusions and Outlook

Ornamental plants in urban and peri-urban areas have a wide range of beneficial effects that are beyond the aesthetical values of green areas. The positive role of ornamental plants on urban environments and their effect on the residents is well known. This review

provides information and suggestions concerning the utilization of ornamental species, including herbaceous and woody species, as biological elements that can be exploited for improving the stability of slopes and reducing the risk of landslides. The potential benefits of ornamental plants have been illustrated and the biological traits that could help in improving slope stability have been reported and discussed. In this respect the recent events in Western Europe and North Italy, in regards to flooding and landslides, have shown that slope management is also of paramount importance in countries where these kinds of events used to be rare in the past but are becoming more frequent due to climate change. Under this scenario the application of nature-based and hybrid solution (i.e., nature-based solutions in combination with conventional engineering solutions) approaches for landslide risk management is calling for the attention of researchers. One of the major questions to be answered regards the selection of suitable species. Species with deeper roots are more effective in preventing shallow soil failures, as their roots and stems provide mechanical reinforcement and restraint and their root uptake and foliage interception modify slope hydrology. The length of the establishment period is also important in selecting species since some species have been shown to establish themselves better and faster than others. Further, with regard to woody species in particular, information on root system architecture, root growth rates, and lifespan would provide environmental managers with data which would enable them to more efficiently manage slopes. Unfortunately (and mainly due to the difficulty of carrying out in situ researches that are time and money consuming and not easily replicable), most of this information is not still available, and therefore further investigations are required for providing enough details that could help the ornamental species selection for slope condition areas.

From the analysis of the literature, a need to extend research to other plant species, differing in root architecture, and to comprehend how future results can be applied at practical level is emerging. Although many models have been developed to predict rainfall dynamics and subsequent erosion potential, few of these have taken root architecture and dynamics into account in terms of their ability to improve soil water flow predictions. This review provides useful suggestions and research directions that can be considered for further studies and investigations focusing on ornamental plants as key elements for controlling erosion and increasing slope stability.

**Author Contributions:** Conceptualization, A.F. (Antonio Ferrante) and D.R.; methodology, A.F. (Alessandra Francini); S.T.; D.R.; F.F. and A.F. (Antonio Ferrante); formal analysis, A.F. (Alessandra Francini); S.T.; D.R.; F.F. and A.F. (Antonio Ferrante); writing—original draft preparation, A.F. (Alessandra Francini); S.T.; D.R. and A.F. (Antonio Ferrante); writing—review and editing, A.F. (Alessandra Francini); S.T.; D.R.; F.F. and A.F. (Antonio Ferrante); supervision, D.R.; F.F. and A.F. (Antonio Ferrante). All authors have read and agreed to the published version of the manuscript.

**Funding:** This research received no external funding.

**Conflicts of Interest:** The authors declare no conflict of interest.

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
