# Peer review of "Biological Contribution of Ornamental Plants for Improving Slope Stability along Urban and Suburban Areas"

_horticulturae, doi:10.3390/horticulturae7090310_

Round 1

Reviewer 1 Report

Comments to the Authors:

The manuscript titled “Role of ornamental plant root systems as an erosion control strategy along urban and suburban side slopes” offers review of plant ability to aid in slope stability in urban areas. This role has important function as one of natural based solutions that could be used to improve quality of life of urban dwellers. As slope stability is wide subject it is fair to state that this review if focused on biological method to control slope stability omitting other engineering solutions or combined approaches.

Overall the manuscript is well researched and has abundance of literature but in my opinion it comes with some significant shortcomings. Those would be discussed through general and specific comments.

My concern is that title of this review does not fully match the subject that was reviewed in this submission. Great deal of space has been given to certain aspects which have marginal meaning to presented title. Such examples were root physiology with little on non-reference to performing on sloped sites or in the urban environments. Role of urban soils on slope stability is not even mentioned. Other example is root growth and root:shoot ratio which were heavily referenced with 18 different references. It seems that this part was used to bulk literature list with small contribution to overall body of the manuscript. Literature list lacks recent publications. References for urban ecosystems were scarce.

Also clear statement about objectives of this study and questions that are being addressed is missing. Limitation of this research were not discussed. Conclusions were not supported with presented results. Also important part of review articles is implications for future research and unanswered research questions whose were also missing from this manuscript.  Disadvantages of the use of plants for slope stability were not discussed. What are some negative implications? What about invasiveness of some plants that are recommended for improving slope stability which could spread in urban ecosystems? Use of other techniques for slope stability with plants – bioengineering approach that is not mentioned?

Overall it seems that this manuscript is mix of different parts that were not bled well together and with lack of connection to manuscript title. To conclude, in my opinion this manuscript lacks with focus to presented title and has many items for improvement.

Specific comments with regards to text lines:

There are some minor changes that should be done in text which is mostly regarding use of style issues. For instance, repeating of two same words in same sentence: L135 presence; L139 among, L296-297 should, L299-300 reach, etc. English writing needs some checking by native speaker.

L18-19 extra-urban? This would relate to outside urban settings? If so than it is not relevant to manuscript title and research topic.

L33 functions? Do you mean services? As function is something that plants or green areas do or provide with no regards to human population, and services do improve quality of life for urban dwellers. So term is not right.

L47 insert is before more stable

L49 Citation is missing for factors that can reduce soil erosion

L94 in a sloped soil,

L95-96 unclear sentence

L99-100 unclear sentence. What does must follow mean?

L103-104 style of this sentence is wrong: where are not easy methods

L114 Typo: properties. And it should state texture as that is what text in brackets refers to

L115-116 style of this sentence is wrong as order is not right.

L128 Unclear statement

L128-129 Surplus sentence

L129-130 Unclear sentence

L135 In the presence

L135 due to presence of change to by

L140 Why only at ornamental plants?

L148 sunflower is annual and not perennial plant

L165 citation is missing for this sentence?

L204, 210, 416,433, 443 Italic?

L251-258 Why did you not mention shot term waterlogging that is often in urban areas due to high amount of rainfall in short periods?

270 oxygen or subscript

L276-280 It seems that this sentences would fit with L155 or L189 paragraphs.

L284-287 same comment as previous

L296-297 Style of sentence. Should is mentioned two times.

L331-335 This has very little reference to manuscript topic as those were papers about nursery production improvements with regard to new roots production with cuttings treatments.

L365 kinetic energy of raindrop is already mentioned at L62

L410 Unclear sentence??

Author Response

Comments to the Authors:

The manuscript titled “Role of ornamental plant root systems as an erosion control strategy along urban and suburban side slopes” offers review of plant ability to aid in slope stability in urban areas. This role has important function as one of natural based solutions that could be used to improve quality of life of urban dwellers. As slope stability is wide subject it is fair to state that this review if focused on biological method to control slope stability omitting other engineering solutions or combined approaches.

Overall the manuscript is well researched and has abundance of literature but in my opinion it comes with some significant shortcomings. Those would be discussed through general and specific comments.

A.A. the manuscript has been revised and improved considering the reviewer’s comments. The current review is intentionally focused on the biological contribution of the slope stability. The engineering solutions are not reported in this work since the aim of the review was to highlight the potential role and contribution of the ornamental plants in the slope stability. At practical level this information can be useful when integrated with engineering solutions. Nevertheless, some information about biomechanical characteristics of root systems has been added [Moresi, F. V., Maesano, M., Matteucci, G., Romagnoli, M., Sidle, R. C., & Scarascia Mugnozza, G. (2019). Root biomechanical traits in a montane Mediterranean forest watershed: variations with species diversity and soil depth. Forests, 10(4), 341; Tsige, D., Senadheera, S., & Talema, A. (2020). Stability analysis of plant-root-reinforced shallow slopes along mountainous road corridors based on numerical modeling. Geosciences, 10(1), 19].

My concern is that title of this review does not fully match the subject that was reviewed in this submission. Great deal of space has been given to certain aspects which have marginal meaning to presented title. Such examples were root physiology with little on non-reference to performing on sloped sites or in the urban environments. Role of urban soils on slope stability is not even mentioned. Other example is root growth and root:shoot ratio which were heavily referenced with 18 different references. It seems that this part was used to bulk literature list with small contribution to overall body of the manuscript. Literature list lacks recent publications. References for urban ecosystems were scarce.

A.A. As specified in the comments above, the work was focused on the biological role of plants with their roots system on the improving the stability of slopes around the urban areas. Title has been changed focusing the work done in this prospective. The lack of reference is due to the lack of specific papers on root physiology in slope conditions in urban environment. This has been specified in the text because it can be research fields to be investigated. On the contrary, a wide number of publications is available on the root:shoot ratio and in this review the roots system is considered particularly important for increasing slope stability and reduce soil erosion. According to the reviewer’s comments the references on root:shoot have been revised and the urban ecosystem have been revised and additional references have been included.

Also, clear statement about objectives of this study and questions that are being addressed is missing. Limitation of this research were not discussed. Conclusions were not supported with presented results. Also, important part of review articles is implications for future research and unanswered research questions whose were also missing from this manuscript.  Disadvantages of the use of plants for slope stability were not discussed. What are some negative implications? What about invasiveness of some plants that are recommended for improving slope stability which could spread in urban ecosystems? Use of other techniques for slope stability with plants – bioengineering approach that is not mentioned?

A.A. thank you for the comments, objectives and questions have been revised and improved. Limitations have been highlighted in the text with appropriate authors’ discussion. Conclusions and been revised and the text that is not supported by results has been suggested as potential further investigations. Disadvantages have been included in the limitations as well as the negative implications. The invasiveness of some plants was not considered in the work because it was out of our scope, however, this important aspect has been mentioned in the manuscript.

Overall it seems that this manuscript is mix of different parts that were not bled well together and with lack of connection to manuscript title. To conclude, in my opinion this manuscript lacks with focus to presented title and has many items for improvement.

A.A. All the manuscript has been revised considering the comments of both reviewers. Title has been changed considering the focus of the work on the biological role of ornamental plants for increasing stability of the slopes in urban and suburban areas.

Specific comments with regards to text lines:

There are some minor changes that should be done in text which is mostly regarding use of style issues. For instance, repeating of two same words in same sentence: L135 presence; L139 among, L296-297 should, L299-300 reach, etc. English writing needs some checking by native speaker.

A.A. thank you for the comments, the sentences have been revised and English has been revised in all the manuscript.

L18-19 extra-urban? This would relate to outside urban settings? If so than it is not relevant to manuscript title and research topic.

A.A. Thank you for the comments, since the manuscript has been focused on urban environments the term extra-urban has been removed from the text.

L33 functions? Do you mean services? As function is something that plants or green areas do or provide with no regards to human population, and services do improve quality of life for urban dwellers. So term is not right.

A.A. Done. The ecosystem services of plants in urban environment have been reported in the manuscript as introduction of the work.

L47 insert is before more stable

A.A. Done

L49 Citation is missing for factors that can reduce soil erosion

A.A. Appropriate citation has been added: Pickup, G., & Marks, A. (2000). The Journal of the British Geomorphological Research Group, 25(5), 535-557. Lopez-Vincente et al., 2021. Journal of Environmental Management, 278, 111510.

L94 in a sloped soil,

A.A. Done

L95-96 unclear sentence

A.A. revised

L99-100 unclear sentence. What does must follow mean?

A.A. the text has been revised

L103-104 style of this sentence is wrong: where are not easy methods

A.A. text has been revised

L114 Typo: properties. And it should state texture as that is what text in brackets refers to

A.A. text has been revised

L115-116 style of this sentence is wrong as order is not right.

A.A. the sentence has been revised

L128 Unclear statement

A.A. the statement has been revised

L128-129 Surplus sentence

A.A. the sentence has been removed

L129-130 Unclear sentence

A.A.. the sentence has been revised

L135 In the presence

A.A. Done

L135 due to presence of change to by

A.A. Done

L140 Why only at ornamental plants?

A.A. The sentence has been modified.

L148 sunflower is annual and not perennial plant

A.A. yes; it is correct; in the sentence are cited all annual plants (or they are in the Mediterranean climate).

L165 citation is missing for this sentence?

A.A. The citation has been added.

L204, 210, 416,433, 443 Italic?

A.A. Done

L251-258 Why did you not mention shot term waterlogging that is often in urban areas due to high amount of rainfall in short periods?

A.A. the text has been revised.

270 oxygen or subscript

A.A. Done

L276-280 It seems that this sentences would fit with L155 or L189 paragraphs.

L284-287 same comment as previous

A.A. The two paragraphs have been modified.

L296-297 Style of sentence. Should is mentioned two times.

A.A. the text has been revised.

L331-335 This has very little reference to manuscript topic as those were papers about nursery production improvements with regard to new roots production with cuttings treatments.

A.A. The nursery production was not in the aim of the review; nevertheless, some information has been added about the influence of the nursery treatment on root system (Franco, J. A., Cros, V., Bañón, S., González, A., & Abrisqueta, J. M. (2002). Effects of nursery irrigation on postplanting root dynamics of Lotus creticus in semiarid field conditions. HortScience, 37(3), 525-528.; Green, J. J., Baddeley, J. A., Cortina, J., & Watson, C. A. (2005). Root development in the Mediterranean shrub Pistacia lentiscus as affected by nursery treatments. Journal of Arid Environments, 61(1), 1-12.).

L365 kinetic energy of raindrop is already mentioned at L62

A.A. the sentence has been removed

L410 Unclear sentence??

A.A. Sorry for the mistake; the question mark is deleted

Reviewer 2 Report

My suggestions are marked in the manuscript

Author Response

Dear reviewer, we thank you for your observations and suggestions, which we have incorporated in the revised paper. To simplify the identification of the changes, we have highlighted them in yellow, while below you have an indication of the point-by-point responses to your observations in the text.

Line 46: is

A.A.: done

Line 49: reduce                affect

A.A.: done

Lines 61-62: contemporarily       simultaneously

A.A.: done

Line 70: is this discussion valid for all types of climate? if the synthesis refers mainly to the Mediterranean and sub-Mediterranean area, this fact should be specified

A.A.: the worst season, i.e. the winter, is not related only to the Mediterranean and sub-Mediterranean area, for this reason we did not further specify the sentence.

Line 100: this scheme is original?

A.A.: yes; the information is added

Line 101: primary             primarily

A.A.: done

Line 139: it seems to be repetitive

A.A.: Done

Line 140: just for the ornamental ones?

A.A.: The sentence has been modified.

Line 141: active                activate?

A.A.: done

Line 148: sunflower        I suggest to use also the scientific name of the species

A.A.: done

Line 149: eliminate “the”

A.A.: done

Lines 149-156: maybe it would be more appropriate reversing these two paragraphs?

A.A.: Thanks for the suggestion; we reversed the two paragraphs

Line 180: Viburnum        Viburnum sp.

A.A.: The species is Viburnum tinus L.

Line 182: is this statement valid for all Viburnum species?

A.A.: Only for Viburnum tinus L.; the sentence is modified.

Line 197: Callistemon and Viburnum highlighted

A.A.: we corrected the scientific name of Callistemon and Viburnum.

Line 204: Callistemon highlighted

A.A.: done

Line 210: Arabidopsis highlighted

A.A.: done

Line 230: geranium and impatiens highlighted

A.A.: we added the scientific name

Lines 299-300: I think this phrase needs to be improved

A.A.: The phrase has changed according to your suggestion.

Line 318: Eucaliptus highlighted

A.A.: we added the scientific name

Line 331: you must re-check the journal's requirements for writing scientific names. Is it necessary to write the names of the authors (as in this case)? Then it must be applied to each species, at the first appearance in the text

A.A. According to your suggestion and journal’s requirements, we reported the scientific names for all cited species.

Line 334: Camelia highlighted

A.A.: done

Line 410: assure?

A.A. Sorry for the mistake; the question mark was removed

Line 416-417: Stipa e Rosmarinus highlighted

A.A.: done

Line 488: effective at preventing             effective in preventing

A.A.: done

Line 495: literature         I suggest to delete         research needs                               research directions

A.A.: done

Round 2

Reviewer 1 Report

It is commendable that authors had revised the title of the manuscript. Revised title better describes the content of the article. Also it is positive that objectives were included in the manuscript, but research limitations and research gaps were scarcely discussed. In my opinion, as this is categorized as review article, requires further emphasis. This can be done in separate paragraph in results or in a discussion. It also seems, from the cited literature that lots of evidence reports and research were of Mediterranean based species which is also one thing to highlight. This could be seen not only as limitation of this study, but also as an opportunity to increase citation and visibility of the article.

Overall changes were made to the manuscript, which resulted with sufficient improvement. Style of the manuscript has also been improved which makes it easier to readers to follow. But beside before mentioned general comments, there were some specific minor changes that would be required before acceptance of this manuscript:

L75 Typo: pant → plant

L87 style in which this sentence starts is wrong. It should not start with of course, as this is general statement that doesn’t need further emphasis.

L131 Figure 2. Please state what does this line left from tree on A section of this figure represent?

L470 Table 1. Typo: Hippophoae →   Hippophae

Author Response

Reviewer 1

It is commendable that authors had revised the title of the manuscript. Revised title better describes the content of the article. Also it is positive that objectives were included in the manuscript, but research limitations and research gaps were scarcely discussed. In my opinion, as this is categorized as review article, requires further emphasis. This can be done in separate paragraph in results or in a discussion. It also seems, from the cited literature that lots of evidence reports and research were of Mediterranean based species which is also one thing to highlight. This could be seen not only as limitation of this study, but also as an opportunity to increase citation and visibility of the article.

A.A. thank you for the further comments and suggestions. A dedicated paragraph has been reported on the limitations and research gaps. The abiotic stresses cited can be observed in Mediterranean countries; therefore, we added the cold or freezing stress as additional stress that along with flooding are typical of Northern Countries. Less information is available on the effect of cold on the performance of plants as slope stabilizers.

Overall changes were made to the manuscript, which resulted with sufficient improvement. Style of the manuscript has also been improved which makes it easier to readers to follow. But beside before mentioned general comments, there were some specific minor changes that would be required before acceptance of this manuscript:

A.A. all the manuscript has been revised and improved. We hope that it is now acceptable for publication.

L75 Typo: pant → plant

A.A. Done

L87 style in which this sentence starts is wrong. It should not start with of course, as this is general statement that doesn’t need further emphasis.

A.A. The statement has been removed from the text.

L131 Figure 2. Please state what does this line left from tree on A section of this figure represent?

A.A. the legend of figure has been improved. The line represents the plant growth response to the gravity. In slope conditions the plants try to grow in a vertical position reducing the angle between the ax of trunk and the soil in slope, as well as occurs underground to the roots.

L470 Table 1. Typo: Hippophoae →   Hippophae

A.A. Done; sorry for the mistake.

Reviewer 2 Report

Please see the comments in the manuscript

Author Response

Dear reviewer, we thank you for your observations and suggestions, which we have incorporated in the revised paper. Below you have indication of the point-by-point responses to your observations in the text.

General observation: I think that some phrases need to be checked for English

A.A.: all the manuscript has been revised and improved also for English. We hope that it is now acceptable for publication.

Line 58: properties?

A.A.: done

Line 62: destruction?

A.A.: done

Line 75: pant      plant

A.A.: done

Line: 75: I think citation is needed

A.A.: Appropriate citation has been added: Ghestem, M.; Cao, K.; Ma, W.; Rowe, N.; Leclerc, R.; Gadenne, C.; Stokes, A. A framework for identifying plant species to be used as ‘ecological engineers’ for fixing soil on unstable slopes. PloS one 2014, 9(8), e95876. https://doi.org/10.1371/journal.pone.0095876

Lines 87-95: in this paragraph is about biochemical or maybe biophysical or physical (mechanical) properties?

A.A.: Thanks for the suggestion: biophysical word is used.

Line 97: The concept of green management seems to be somehow different. Green space management may be more appropriate

A.A.: done

Line 97: Therefore, the species                                the species must be selected considering....

A.A.: done

Line 99: citation?

A.A.: Appropriate citation has been added: Ghafari, S.; Kaviani, B.; Sedaghathoor, S.; Allahyari, M. S. Ecological potentials of trees, shrubs and hedge species for urban green spaces by multi criteria decision making. Urban For. Urban Green. 2020, 55, 126824. https://doi.org/10.1016/j.ufug.2020.126824

Line 149: using biological solutions can have several practical applications for slope stabilization?

A.A.: the sentence was modify

Line 149: ,

A.A.: done

Line 150: ,

A.A.: done

Line 154: For rocky slopes...

A.A.: done

Line 158: under slopes                  under different slope conditions

A.A.: done

Line 175: Different abiotic stresses affecting the plant  Different abiotic stresses affect the plants

A.A.: done

Line 207: The plants modify the R/S ratio during drought stress?

A.A.: The sentence was modify.
